# Factors influencing the participation of surrogate decision-makers for advanced cancer patients in advance care planning: A cross-sectional study

Zhihao Han[1,2☯], Jingjing Su[1☯], Guiyue Ma[1], Yong Fang[3], Linxia Tang[1], Xiaoqin Ma [1]*

**1** Department of Nursing, Zhejiang Chinese Medical University, Hangzhou City, Zhejiang Province, China, **2** Zhejiang Chinese Medical University First Clinical School of Medicine, Hangzhou City, Zhejiang Province, China, **3** Department of Nursing, Changsha Medical College, Changsha City, Hunan Province, China

☯ These authors contributed equally to this work.
* 20041028@zcmu.edu.cn

## Abstract

### Aim

The aim of this study was to explore the factors influencing surrogate decision-makers (SDMs) engagement in Advance Care Planning (ACP).

### Design

A cross-sectional Study.

### Method

A total of 285 SDMs of advanced cancer patients were recruited using convenience sampling methods. All eligible participants completed the structured questionnaires. Univariate and correlation analyses were used to explore the relationship between different variables and the ACP engagement of SDMs. Factors significant previously in univariate or correlation analyses were included in the multiple linear regression analyses with $P < 0.05$.

### Results

Eight factors in univariate analyses and three factors in correlation analyses with $P < 0.05$ were taken into multiple linear regression analyses. Finally, seven variables revealed a statistically significant association with ACP engagement of SDMs and were included in the multiple linear regression model.

### Conclusion

This cross-sectional study demonstrated that multiple factors, including experiences of medical decision-making, education level, knowledge of ACP, treatment

**Data availability statement:** All relevant data are within the manuscript and its Supporting Information files.

**Funding:** The author(s) received no specific funding for this work.

**Competing interests:** The authors have declared that no competing interests exist.

expenditure, uncertainty of disease, social support, and life orientation, significantly impacted the engagement of SDMs in ACP.

## Impact

Based on this study, we have identified some factors influencing the ACP engagement of SDMs, which might provide basic information for construction of the educational scheme of ACP in mainland China and help to improve SDMs' engagement in clinical ACP practice in the future.

## Patient or public contribution

SDMs of advanced cancer patients participated in this study, and assisted in assessing the validation of the measurements.

---

## 1. Introduction

Advance Care Planning (ACP) is increasingly implemented in oncology and other fields to plan future healthcare and enhance the alignment of care with patients' goals. In 2017, the European Association for Palliative Care proposed a definition of ACP as a process that "enables individuals to define goals and preferences for future medical treatment and care, to discuss these goals and preferences with family and healthcare providers, and to record and review these preferences if appropriate" [1]. The definition clearly indicates that ACP encompasses far more than merely completing advance directives (ADs), which are instructional directives or "living wills" that enable individuals to make decisions or state goals for future medical treatment or appoint surrogate decision makers (SDMs) to make decisions on their behalf [2]. ADs can be an important component of the ACP process. While ACP is a broader process involving discussions and planning, ADs serve as legal documents that formalize these preferences. The implementation of ADs has been established in many countries and regions, such as England, the United States, Austria, and Germany. However, the practices of ADs were marked by heterogeneity across countries, as shown in a 2021 study, which showed that only 15 of the 28 European Union countries had developed specific rules on ADs, which, when written, are legally binding in 86% of cases [3]. In mainland China, the implementation of ACP is hindered by deeply rooted cultural norms, legal gaps, and structural inequities. Culturally, Confucian familism and collectivist values prioritize familial consensus over individual autonomy, often positioning SDMs as primary arbiters even when patients retain decision-making capacity. This is compounded by societal taboos surrounding open discussions about death and end-of-life care, which are perceived as inauspicious. Legally, the absence of national legislation recognizing advance directives leaves ACP documentation unenforceable. Structurally, urban-rural disparities are stark: tier-1 hospitals pilot innovative solutions like WeChat-based ACP modules to facilitate remote family involvement, while rural areas face fragmented healthcare resources and inadequate clinician training.

## 2. Background

According to the latest data from the International Agency for Research on Cancer in 2022, there were globally close to 20 million new cases of cancer alongside 9.7 million deaths from cancer [4]. China accounted for 24.1% of new cases and 26.8% of cancer-related deaths [4]. Many individuals in China are diagnosed with advanced stages of cancer due to delayed cancer screening and prevention efforts, and they have to face death and life-threatening complications during treatment [5]. Disease progression might result in them suddenly falling into unconsciousness or becoming incapable of expressing their own treatment preferences. Life-sustaining treatments, while potentially life-prolonging, may also lead to significant suffering and diminished quality of life [6]. To maintain individuals' autonomy and protect their right of self-determination, discussions about medical preferences must be conducted when individuals are still capable of decision-making.

ACP has emerged as a solution to this problem, enabling individuals to choose their preferences for medical treatment and care in advance, even before they become incapacitated [7]. ACP also encourages individuals and their families to participate in shared decision-making discussions with healthcare professionals. This collaborative pattern helps family members understand the individual's health status, values, and goals, easing decision-making burdens and preventing conflict. In mainland China, the patient's SDM is typically a family member. The designation of an SDM generally requires the patient to sign relevant documents provided by the hospital and obtain hospital approval.

In Chinese nursing cultures, where interdependence and achieving greater good for the group are prioritized over patients' medical preferences, medical decisions are often influenced by family dynamics. In such contexts, medical decisions are often made collaboratively within the family, with the goal of maintaining family harmony and ensuring the best possible outcomes for all members involved [8]. Filial piety norms, deeply rooted in Confucian traditions, often result in family members hiding the patient's condition in order to protect the patient from psychological distress [8]. This cultural prioritization of familial harmony over patient autonomy creates barriers to ACP initiation. However, shared responsibility among relatives can also foster collaborative planning. When individuals experience cognitive impairment and other conditions of loss of decision-making capacity, family members are often deemed to be the primary decision-makers on their behalf. Therefore, most individuals in China choose family members, such as spouses, children, or parents, as SDMs [9] due to the family concept. Engagement of family members in the ACP process is a critical component in the improvement of ACP practice.

Currently, most studies are mainly focused on individuals' engagement in ACP, such as the completion rate of ACP, or the engagement in the comprehensive ACP process measured by the ACP Engagement Survey [10–13]. However, there are few studies related to ACP involving both SDMs and patients in mainland China [14]. The purpose of this study was to explore the individual influencing factors of the SDMs' engagement in ACP with patients. This study might provide some information for the construction of the educational scheme of ACP in mainland China, particularly for SDMs of advanced cancer patients. The findings could help improve SDMs' engagement in clinical ACP practice in the future by developing targeted educational programs.

## 3. The study

### 3.1. Aim

The aim of this study was to explore the factors influencing of ACP engagement of SDMs.

## 4. Methods

### 4.1. Study design

This study is a cross-sectional study. Data collection was conducted from February 2023 to July 2023 at First Affiliated Hospital of Zhejiang Chinese Medical University.

### 4.2. Participants and selection criteria

Participants were recruited using convenience sampling methods. Participants were invited to take part in this survey by well-trained investigators. The investigators explained in detail the purpose of this study and obtained informed consent from all participants. All eligible participants completed the structured paper questionnaires by themselves or with the assistance of professional investigators if needed.

The inclusion criteria for SDMs of advanced cancer patients were as follows: (1) 18 years old or above; (2) a marital or parental relationship with the advanced tumor patient (such as parents, spouse, children, or sibling) and being appointed as the SDM by the patient through a formal written designation, in accordance with the ethical principles and clinical guidelines relevant to the protection of patients' rights and interests; (3) providing informed consent and cooperation with the survey; (4) possessing the ability to communicate effectively and complete the questionnaire independently.

Based on the principles of sample size estimation for empirical methods [15], this study encompasses 23 variables. Taking 5–10 times the number of variables, the estimated sample size ranges from 115 to 230 participants. To account for a 20% dropout rate, the final sample size range was adjusted to 138–276 participants.

### 4.3. Measurements

To identify the factors influencing the ACP engagement of SDMs of patients with advanced cancer, a structured questionnaire was designed in this study. Based on the results of a previous study [16], factors were categorized into three dimensions, the individual, environmental, and behavioral dimensions. This study explored how individual and environmental factors that influence the ACP engagement of SDMs by the theory of triadic reciprocal determinism. Individual factors such as self-efficacy and outcome expectations mainly reflected in life orientation ability and uncertainty about illness. Environmental factors primarily manifested as the level of social support for SDMs, and behaviors involving the evaluation of SDMs' ACP engagement.

**4.3.1. Socio-demographic characteristics of SDMs and cancer characteristics of patients.** The Socio-demographic and cancer characteristics questionnaire was designed by the researchers. We chose 12 variables in the questionnaire, which might affect the SDMs' ACP engagement in previous studies [9,11,17], including age, sex, education level, marital status, the number of family members of SDMs, annual income, relationship with the advanced cancer patient, classification of patients' cancer, treatment expenditure, patients' activities of daily living (ADL) independence level, experiences of making medical decision and knowledge of ACP.

**4.3.2. Chinese Version of 17-item Advance Care Planning Engagement Survey for Surrogate Decision Makers (ACP-17-SDM-C).** The Chinese version of ACP-17-SDM was translated by Wang Tianhang et al [16] from the English version of ACP-17-SDM [18]. The ACP-17-SDM-C consists of three dimensions (serving as SDM, contemplation, and readiness) with 17 items. Each item is rated with a 5-point Likert scale ranging from 1 (not at all/ never) to 5 (very much/ always). The total score of the ACP-17-SDM-C ranges from 17 to 85 with higher scores indicating higher levels of ACP engagement of SDMs. In this study, the ACP-17-SDM-C demonstrated good reliability, with a Cronbach's α coefficient of 0.915.

**4.3.3. Chinese Version of Mishel Uncertainty in illness Scale-Family Member Form (MUIS-FM-C).** The levels of disease uncertainty in the family members of patients were evaluated with the Chinese version of the MUIS-FM, which was translated and revised by Wenying Wang [19]. The MUIS-FM-C consists of four dimensions (ambiguity, lack of clarity, lack of information, and unpredictability) covering 25 items. The total score of the MUIS-FM-C ranges from 25 to 125 with higher scores indicating higher levels of disease uncertainty of family members. In this study, the MUIS-FM-C demonstrated good reliability, with a Cronbach's α coefficient of 0.887.

**4.3.4. Social support rating scale (SSRS).** The SSRS, revised by Shuiyuan Xiao [20], was used to assess the current level of social support. The scale consists of three dimensions, including subjective social support, objective social

support, and social support use. The total score of the SSRS ranges from 12 to 66 with higher scores indicating higher levels of social support. In this study, the SSRS demonstrated good reliability, with a Cronbach's α coefficient of 0.788.

   **4.3.5 Chinese version of Life orientation test-Revised (LOT-R-C).** The Chinese version of LOT-R was translated by Wengen Deng [21] from the English version of LOT-R [22]. The LOT-R-C was mainly used to assess the disposition of optimism or pessimism. The LOT-R-C is a single-dimensional self-assessment scale consisting of 10 items. Each item is rated with a 5-point Likert scale ranging from 0 (strongly disagree) to 4 (strongly agree). The total score of the LOT-R-C ranges from 0 to 24 with higher scores indicating higher levels of optimism. In this study, the LOT-R-C demonstrated good reliability, with a Cronbach's α coefficient of 0.831.

## 4.4. Statistical analysis

The Statistical Package for the Social Sciences (SPSS) software version 26.0 was used for data input and statistical analyses. The scores of questionnaires were summarized using mean±standard deviation for normally distributed data or median (Interquartile Range) for non-normally distributed data. Univariate analyses were used to explore differences in ACP engagement of SDMs across different categories of socio-demographic characteristics. For normally distributed data, the independent samples t-test or ANOVA test was used depending on the number of groups; for non-normally distributed data, the Mann-Whitney U rank-sum test or Kruskal-Wallis H rank-sum test was used accordingly. Pearson and Spearman correlation analyses were calculated to assess the relationships between ACP engagement of SDMs and other variables, such as the uncertainty of disease, social support, and life orientation of the participants. Factors significant previously in univariate or correlation analyses were included in multiple linear regression analyses, with variables removed if P>0.05. Participants with missing data were excluded from the analyses.

## 4.5. Ethical considerations

This study was approved by the Zhejiang Chinese Medical University Ethics Committee on February 16, 2023 (20230216−5). After receiving a full explanation of the study, all participants voluntarily agreed to participate and provided written informed consent before the interviews.

## 5. Results

In this study, of the 285 questionnaires administered, 9 responses were excluded due to missing data or being illegible. Thus, 276 responses were finally analyzed (effective response rate: 96.8%).

## 5.1. Socio-demographic characteristics of SDMs and cancer characteristics of patients

The socio-demographic characteristics of SDMs and cancer characteristics of patients were presented in Table 1. The participants were evenly distributed by age and sex. Most of the participants were married and lived with their family members. Additionally, more than two-thirds of the participants had no experience in medical decision-making (70.3%) or lacked knowledge of ACP (68.5%). Most of the SDMs were the children (44.9%) or spouses (43.8%) of advanced cancer patients. Only 8.7% of patients were fully independent in daily life, while the majority (91.3%) required some level of assistance.

## 5.2. Factors associated with the ACP engagement of SDMs

Based on univariate analyses, the factors significantly influencing the ACP engagement of SDMs were age, sex, education level, experiences of making medical decision, knowledge of ACP, treatment expenditure, annual income, and relationship with the advanced cancer patient. Meanwhile, there was no significant difference in ACP engagement of SDMs categorized by marital status, the number of family members of SDMs, classification of patients' cancer and patient's

**Table 1. Socio-demographic characteristics of SDMs and cancer characteristics of patients (N = 276).**

| characteristics | | Frequency (n) | Percentage (%) |
|---|---|---|---|
| **Age (years)** | 18–40 | 98 | 35.5 |
| | 41–60 | 113 | 40.9 |
| | ≥61 | 65 | 23.6 |
| **Sex** | Female | 144 | 52.2 |
| | Male | 132 | 47.8 |
| **Education status** | Primary school or less | 46 | 16.7 |
| | Junior high school | 72 | 26.1 |
| | Senior high school | 57 | 20.7 |
| | College | 90 | 32.6 |
| | Master degree or above | 11 | 4.0 |
| **Marital status** | Unmarried or widowed or divorced | 37 | 13.4 |
| | Married | 239 | 86.6 |
| **Experiences of making medical decision** | No | 194 | 70.3 |
| | Yes | 82 | 29.7 |
| **Knowledge of ACP** | Unknown | 189 | 68.5 |
| | Heard of ACP | 55 | 19.9 |
| | Partly known | 28 | 10.1 |
| | Known | 4 | 1.4 |
| **Treatment expenditure** | ≤ ¥10,000 (USD$1370) | 110 | 39.9 |
| | ≤ ¥30,000 (USD$4110) | 120 | 43.5 |
| | ≤ ¥50,000 (USD$6850) | 28 | 10.1 |
| | >¥50,000 (USD$6850) | 18 | 6.5 |
| **Annual income** | ≤ ¥10,000 (USD$1370) | 112 | 40.6 |
| | ≤ ¥30,000 (USD$4110) | 101 | 36.6 |
| | ≤ ¥50,000 (USD$6850) | 30 | 10.9 |
| | >¥50,000 (USD$6850) | 33 | 12.0 |
| **The number of family members of SDMs** | ≤3 | 110 | 39.9 |
| | 4-5 | 121 | 43.8 |
| | ≥6 | 45 | 16.3 |
| **Relationship with the advanced cancer patient** | Children | 124 | 44.9 |
| | Parents | 19 | 6.9 |
| | Spouse | 121 | 43.8 |
| | Siblings | 6 | 2.2 |
| | Grandchildren | 6 | 2.2 |
| **Classification of patients' cancer** | Digestive system tumors | 112 | 40.6 |
| | Hematologic cancers | 34 | 12.3 |
| | Genitourinary cancer | 28 | 10.1 |
| | Head and neck cancer | 25 | 9.1 |
| | Thoracic cancer | 77 | 27.9 |
| **Patient's ADL* independence level** | Fully independent | 24 | 8.7 |
| | Partially dependent# | 77 | 27.9 |
| | Substantially dependent# | 62 | 22.5 |
| | Completely dependent# | 113 | 40.9 |

*ADL: Activities of daily living. #dependent: The categories of "partially dependent, substantially dependent, completely dependent" were established to reflect gradations in care needs, with mutually exclusive definitions that align with the Barthel Index validated in Chinese populations.

independence level (Table 2). In addition, the correlation between the uncertainty of disease, social support, life orientation of participants with ACP engagement of SDMs were shown in Table 3. The ACP engagement of SDMs was negatively associated with uncertainty of disease (P<0.01) and positively associated with social support and life orientation of participants (both with P<0.01).

### 5.3. Multiple linear regression analyses for factors associated with the ACP engagement of SDMs

Eight factors in univariate analyses and three factors in correlation analysis with P<0.05 were included in multiple linear regression analyses. The stepwise regression method was used to assess the relationship among the variables. All independent variables were shown in Table 4. Tolerance (TOL) of each variable in the model was greater than 0.1, and the variance inflation factor (VIF) was less than 5. Therefore, there was no multicollinearity among the variables.

Finally, as shown in Table 5, only seven variables were found to have statistically significant associations with ACP engagement among SDMs. These seven variables, included in the multiple linear regression model, explained 56.7% of the total variation (Adjusted $R^2$=0.567; F=52.448; P<0.001). Among those variables, all except the uncertainty of disease showed a strong positive association with ACP engagement among SDMs.

## 6. Discussion

This study investigated the engagement of SDMs in ACP for Chinese advanced cancer patients, which provided valuable insights for the implementation of ACP in mainland China. To our knowledge, this study was the first large-sample study focusing on the factors influencing ACP engagement of SDMs for advanced cancer patients in mainland China. Currently, there are few studies related to ACP engagement of patients at the end of life in mainland China due to traditional Chinese culture, which limits the spread and implementation of ACP [17].

A total of 276 SDMs of advanced cancer patients were analyzed in this study, with a balanced distribution of sex (52.2% female and 47.8% male). Most participants were children or spouses of advanced cancer patients. Among the participants, SDMs younger than 60 years old accounted for a large proportion, which was similar to the findings of Wang's study [17]. Considering the surrogate relationship with patients, we considered the following possibilities of this phenomenon. On one hand, most advanced cancer patients were older people, and their children often assumed the role of SDMs. On the other hand, for some middle-aged cancer patients, their spouse might become the first choice of SDM because their children were too young and their parents were too old to make decisions. In our study, participants aged 18–40 years exhibited the highest level of ACP engagement among the different age groups, which contrasts with Liu's findings that older SDMs had higher ACP engagement [23]. This discrepancy may be attributed to differences in the characteristics of the care recipients in our respective studies. Specifically, Liu's study focused on community-dwelling older patients with chronic diseases, whereas our study centered on SDMs of advanced cancer patients. The reason why participants aged 18–40 years are more actively involved in ACP for patients may be that younger SDMs are generally more adaptable and have stronger capabilities in accessing ACP-related resources [17].

Experiences of medical decision-making were significantly associated with ACP engagement, and participants with experiences showed more positive ACP engagement. However, only 29.7% of the participants had experiences of medical decision-making in this study, which might be attributed to the limited implementation of the doctor-patient-surrogate decision-making model in mainland China. This limited implementation may hinder SDMs from fully obtaining the patients' medical information [24–26]. A recent study showed that SDMs with experiences of decision-making tend to be more deliberate when making medical decisions due to the knowledge of disease trajectory [27].

The univariate analyses showed that education level was positively correlated with ACP engagement. This result was similar to the study of Keam et al [28], which also showed that participants with a higher education level exhibited more positive ACP engagement than those with a lower education level. Participants with a high education level tended to have a strong sense of autonomy and to consider the value of life and future medical intentions when actively seeking disease

**Table 2. The difference of ACP engagement of SDMs with different socio-demographic and care recipients characteristics.**

| Characteristics | Subscales | | | Total [mean±SD]/ median(IQR)] |
|---|---|---|---|---|
| | Serving as SDM [mean±SD]/ median(IQR)] | Contemplation [mean±SD]/ median(IQR)] | Readiness [mean±SD]/ median(IQR)] | |
| **Age (years)** | | | | |
| 18–40 | 25.10±3.63 | 11.00 (7.00) | 19.00 (7.00) | 55.50 (18.00) |
| 41–60 | 23.38±5.25 | 10.00 (7.00) | 17.00 (7.00) | 49.00 (21.00) |
| ≥61 | 21.46±5.76 | 7.00 (8.00) | 14.00 (7.00) | 43.00 (24.00) |
| F/H | 10.996 | 5.605 | 16.536 | 16.604 |
| P | **<0.001** | 0.061 | **<0.001** | **<0.001** |
| **Sex** | | | | |
| Female | 22.51±4.93 | 9.00 (7.00) | 16.00 (8.00) | 47.50 (19.00) |
| Male | 24.67±4.96 | 10.00 (8.00) | 17.50 (9.00) | 54.00 (22.00) |
| t/Z | −3.627 | −0.557 | −2.478 | −2.678 |
| P | **<0.001** | 0.577 | **0.013** | **0.007** |
| **Education level** | | | | |
| Primary school or less | 18.85±4.77 | 7.00 (2.00) | 13.00 (3.00) | 40.74±9.02 |
| Junior high school | 21.38±4.64 | 7.00 (4.00) | 15.00 (5.00) | 45.33±10.73 |
| Senior high school | 24.93±3.63 | 12.00 (6.00) | 19.05±4.38 | 55.44±10.61 |
| College | 27.00 (5.00) | 13.00 (7.00) | 20.00 (8.00) | 57.83±10.55 |
| Master degree or above | 29.00 (2.00) | 16.00 (5.00) | 22.36±3.53 | 65.27±8.75 |
| F/H | 92.646 | 59.416 | 77.940 | 34.457 |
| P | **<0.001** | **<0.001** | **<0.001** | **<0.001** |
| **Marital status** | | | | |
| Unmarried or widowed or divorced | 23.11±4.60 | 10.19±3.95 | 17.00 (9.00) | 48.00 (22.00) |
| Married | 24.00 (8.00) | 10.00 (8.00) | 17.00 (7.00) | 49.00 (21.00) |
| Z | −0.831 | −0.161 | −0.284 | −0.311 |
| P | 0.406 | 0.872 | 0.776 | 0.756 |
| **Experiences of making medical decision** | | | | |
| No | 23.00 (6.00) | 8.00 (6.00) | 15.00 (6.00) | 46.00 (15.00) |
| Yes | 28.00 (4.00) | 14.00 (5.00) | 21.00±4.45 | 64.00 (12.00) |
| Z | −7.346 | −6.780 | −7.294 | −7.852 |
| P | **<0.001** | **<0.001** | **<0.001** | **<0.001** |
| **Knowledge of ACP** | | | | |
| Unknown | 21.90±4.78 | 8.00 (6.00) | 15.90±4.13 | 46.00 (15.00) |
| Heard of ACP | 28.00 (3.00) | 14.00 (5.00) | 21.25±3.94 | 64.00 (11.00) |
| Partly known | 27.36±3.47 | 13.61±3.34 | 22.14±4.08 | 63.11±9.15 |
| Known | 26.50±4.93 | 10.75±4.27 | 17.50±5.80 | 54.75±14.18 |
| F/H | 70.245 | 59.818 | 36.639 | 82.403 |
| P | **<0.001** | **<0.001** | **<0.001** | **<0.001** |
| **Treatment expenditure** | | | | |
| ≤ ¥10,000 (USD$1370) | 22.40±4.52 | 7.00 (6.00) | 15.00 (5.00) | 43.00 (14.00) |
| ≤ ¥30,000 (USD$4110) | 25.00 (7.00) | 11.50 (7.00) | 18.50 (8.00) | 53.72±12.98 |
| ≤ ¥50,000 (USD$6850) | 25.50 (7.00) | 12.00 (7.00) | 19.18±4.91 | 54.36±11.71 |
| >¥50,000 (USD$6850) | 28.50 (5.00) | 12.17±3.20 | 20.56±4.25 | 63.00 (12.00) |
| H | 20.812 | 28.671 | 27.781 | 28.729 |

*(Continued)*

**Table 2.** (Continued)

| Characteristics | Subscales | | | Total [mean±SD]/ median(IQR)] |
| --- | --- | --- | --- | --- |
| | Serving as SDM [mean±SD]/ median(IQR)] | Contemplation [mean±SD]/ median(IQR)] | Readiness [mean±SD]/ median(IQR)] | |
| P | **<0.001** | **<0.001** | **<0.001** | **<0.001** |
| **Annual income** | | | | |
| ≤ ¥10,000 (USD$1370) | 21.37±4.94 | 7.50(6.00) | 14.50 (5.00) | 43.50 (16.00) |
| ≤ ¥30,000 (USD$4110) | 24.00 (6.00) | 11.00 (7.00) | 18.00 (8.00) | 52.00 (21.00) |
| ≤ ¥50,000 (USD$6850) | 27.50 (5.00) | 12.50 (6.00) | 18.40±4.60 | 55.70±12.45 |
| >¥50,000 (USD$6850) | 27.00 (3.00) | 12.09±3.31 | 20.52±3.95 | 58.97±8.96 |
| H | 42.865 | 23.014 | 37.549 | 40.237 |
| P | **<0.001** | **<0.001** | **<0.001** | **<0.001** |
| **The number of family members of SDMs** | | | | |
| ≤3 | 24.50 (8.00) | 11.50 (8.00) | 18.00 (9.00) | 53.50 (24.00) |
| 4–5 | 23.00 (7.00) | 9.00 (7.00) | 16.00 (7.00) | 47.00 (19.00) |
| ≥6 | 24.16±3.93 | 8.00 (8.00) | 16.00 (7.00) | 48.00 (20.00) |
| H | 4.995 | 5.366 | 4.744 | 5.756 |
| P | 0.082 | 0.068 | 0.093 | 0.056 |
| **Relationship with the advanced cancer patient** | | | | |
| Children | 24.69±4.15 | 11.00 (7.00) | 18.00 (7.00) | 54.00 (19.00) |
| Parents | 21.37±4.11 | 7.74±2.88 | 13.00 (4.00) | 44.21±9.38 |
| Spouse | 22.55±5.77 | 9.00 (8.00) | 15.00 (8.00) | 47.00 (24.00) |
| Siblings | 26.17±2.93 | 10.50±4.46 | 18.83±4.79 | 55.50±11.36 |
| Grandchildren | 23.83±5.19 | 16.00 (8.00) | 19.33±4.80 | 56.50±12.36 |
| F/H | 4.253 | 11.811 | 16.576 | 15.189 |
| P | **0.002** | **0.019** | **0.002** | **0.004** |
| **Classification of patients' cancer** | | | | |
| Digestive system tumors | 23.00 (8.00) | 10.00 (8.00) | 17.00 (8.00) | 51.38±12.55 |
| Hematologic cancers | 24.53±3.59 | 9.49±3.78 | 16.00 (7.00) | 49.00 (19.00) |
| Genitourinary cancer | 23.64±4.65 | 10.50 (9.00) | 17.00 (11.00) | 52.00 (27.00) |
| Head and neck cancer | 23.64±5.83 | 8.00 (9.00) | 16.36±4.89 | 46.00 (27.00) |
| Thoracic cancer | 24.00 (9.00) | 10.00 (7.00) | 17.00 (9.00) | 51.00 (24.00) |
| H | 1.371 | 2.139 | 2.301 | 0.697 |
| P | 0.849 | 0.710 | 0.681 | 0.952 |
| **Patients' ADL independence level** | | | | |
| Fully independent | 25.50 (14.00) | 12.50 (10.00) | 17.46±5.70 | 51.25±16.50 |
| Partially dependent | 26.00 (9.00) | 12.00 (8.00) | 20.00 (10.00) | 60.00 (23.00) |
| Substantially dependent | 23.42±4.74 | 10.00 (5.00) | 16.00 (8.00) | 48.00 (19.00) |
| Completely dependent | 23.00 (6.00) | 8.00 (7.00) | 16.00 (6.00) | 47.00 (15.00) |
| H | 4.265 | 8.211 | 9.602 | 7.006 |
| P | 0.234 | **0.042** | **0.022** | 0.072 |

ADL: Activities of daily living; F: F value of ANOVA test, H: H value of Kruskal-Wallis H rank-sum test; t: t value of independent samples t-test; Z: Z value of Mann-Whitney U rank-sum test; P-values indicating significance at the 0.05 level are shown in bold.

**Table 3. The correlation between the uncertainty of disease, social support, life orientation of participants with ACP engagement of SDMs.**

| Characteristics | Subscales | | | Total |
|---|---|---|---|---|
| | Serving as SDM | Contemplation | Readiness | |
| Uncertainty of disease | −0.485** | −0.452** | −0.469** | −0.524** |
| Social support | 0.514** | 0.410** | 0.501** | 0.534** |
| Life orientation | 0.356** | 0.387** | 0.401** | 0.424** |

**: $P < 0.01$

**Table 4. Independent variables assignment in multiple linear regression analyses.**

| Independent variable | Assignment |
|---|---|
| Age | 1 = 18–40, 2 = 41–60, 3 = ≥61 |
| Sex | 1 = Female, 2 = Male |
| Education level | 1 = Primary school or less, 2 = Junior high school, 3 = Senior high school, 4 = College, 5 = Master degree or above |
| Experiences of medical decision-making | 1 = No, 2 = Yes |
| Knowledge of ACP | 1 = Unknown, 2 = Heard of ACP, 3 = Partly known, 4 = Known |
| Treatment expenditure | 1=≤¥10,000 (USD$1370), 2=≤¥30,000 (USD$4110), 3=≤¥50,000 (USD$6850), 4=>¥50,000 (USD$6850) |
| Annual income | 1=≤¥10,000 (USD$1370), 2=≤¥30,000 (USD$4110), 3=≤¥50,000 (USD$6850), 4=>¥50,000 (USD$6850) |
| Relationship with the advanced cancer patient | 1 = Children, 2 = Parents, 3 = Spouse, 4 = Siblings, 5 = Grandchildren |
| The uncertainty of disease | null |
| The social support | null |
| The life orientation | null |

information [29,30]. Conversely, participants with a low education level might lack knowledge to support their thinking, although they want to express their wishes [31]. Participants with knowledge of ACP exhibited a more positive attitude than those without such knowledge, which is consistent with the study of Park et al [32]. A comprehensive understanding of relatively unfamiliar healthcare concepts like ACP—particularly in cultural contexts where end-of-life care discussions are not yet normalized—is a prerequisite for meaningful participation. Therefore, policymakers, healthcare institutions, and public health advocates should collaborate to develop strategies that integrate ACP education into mainstream healthcare systems, leverage media platforms for awareness campaigns, and ensure accessible resources for SDMs [33]. To enhance the engagement of SDMs with inadequate education in ACP discussions, it is essential to use plain and straightforward language to explain the importance of ACP and to provide them with easily digestible written resources and visual aids. Moreover, offering training programs and securing support from healthcare professionals can also help SDMs better understand the patients' requirements and preferences, thereby enhancing their active participation in ACP discussions.

This study revealed that the treatment cost burden significantly influenced medical decisions made by SDMs for advanced cancer patients. High treatment costs placed substantial economic pressure on SDMs [34,35], which may lead them to prioritize cost-effective care options when patients' preferences are unknown or undocumented. This decision-making process can be influenced by the advanced stage of the disease, which often requires more intensive and costly nursing care [36]. Importantly, this economic strain highlights the urgency of integrating patients into ACP discussions early, ensuring their values and treatment priorities are explicitly documented. The treatment expenditure was comparable to annual income, which indicates that the treatment costs for advanced cancer occupied most of the annual income, imposing a substantial economic burden on their families [37–39]. It is recommended that medical staff

**Table 5. Multiple linear regression analyses of independent variables in ACP engagement of SDMs.**

| Variables | Regression coefficient | Standardized coefficient | t | P | TOL | VIF |
|---|---|---|---|---|---|---|
| Experiences of medical decision-making | 6.144 | 0.224 | 4.934 | **<0.001** | 0.771 | 1.297 |
| Education level | 2.460 | 0.230 | 4.650 | **<0.001** | 0.491 | 2.035 |
| Knowledge of ACP | 1.989 | 0.116 | 2.387 | **0.018** | 0.649 | 1.541 |
| Treatment expenditure | 1.541 | 0.105 | 2.554 | **0.011** | 0.908 | 1.101 |
| Uncertainty of disease | −0.196 | −0.232 | −5.051 | **<0.001** | 0.782 | 1.279 |
| Social support | 0.249 | 0.159 | 3.186 | **0.002** | 0.666 | 1.501 |
| Life orientation | 0.274 | 0.105 | 2.279 | **0.023** | 0.773 | 1.294 |

TOL: tolerance; VIF: variance inflation factor; P-values indicating significance at the 0.05 level are shown in bold.

comprehensively consider the treatment needs and economic status of patients and their families, assisting them in weighing the pros and cons of different treatment options and providing personalized medical decision-making recommendations [40].

This study showed that ACP engagement was negatively correlated with the SDMs' sense of disease uncertainty and positively correlated with the SDMs' social support and life orientation ability. Firstly, according to the Uncertainty in Illness Theory, uncertainty arises when individuals are unable to establish an adaptive cognitive framework for related events [41]. Doctors and healthcare professionals play a vital role in facilitating communication between patients and SDMs by offering support and guidance throughout these stages, especially for SDMs who have no medical knowledge. Secondly, social support refers to the material and spiritual support that individuals obtain from family and other social networks [42,43]. There is a study that reached similar conclusion to this study: participants with high social support showed more active involvement in ACP and higher quality ACP completion [44]. Thirdly, many studies have shown that individuals in a good emotional state tend to exhibit more altruistic behaviors [45–47]. Optimistic SDMs often actively seek help from others to find solutions when encountering difficulties. Therefore, SDMs with strong life orientation ability are more willing to participate in discussions about patients' treatment decisions and exhibit a more positive attitude toward ACP.

Patients play a central role in the development and implementation of the entire ACP, and it is crucial to facilitate communication between patients and SDMs. Firstly, healthcare providers should organize family meetings involving cancer patients and their SDMs. These meetings ensure that patients' values are respected and that SDMs understand the patients' treatment preferences, enabling them to accurately convey these preferences to doctors and nurses [48,49]. Secondly, researchers could develop ACP decision aids tailored to Chinese culture. These tools should primarily focus on patient-related information, helping patients understand their disease characteristics and predict treatment outcomes [49]. Thirdly, hospital administrators should adopt policies aligned with the "Healthy China 2030" initiative, such as including ACP documentation rates in healthcare quality evaluations.

Future practice should focus on providing enhanced training for healthcare providers in effective communication with SDMs and patients about ACP, especially for SDMs and patients unfamiliar with ACP. This is crucial because a lack of knowledge about ACP can significantly hinder the engagement of patients and their SDMs in the ACP process. This includes training on cultural sensitivity, communication techniques, and strategies to overcome barriers to ACP discussions [50]. ACP should be integrated into routine care for advanced cancer patients. Healthcare providers should make it a standard practice to discuss ACP with patients and their SDMs early in the disease trajectory. This can help ensure that patients' preferences and values are respected and that their care aligns with their wishes. There is a need for more intervention studies to test the effectiveness of different ACP strategies.

The limitations of this study should be acknowledged. Recruiting a convenience sample from a single cancer center may limit generalizability to national target populations. Further studies with large, nationally representative samples are needed to substantiate our findings.

## 7. Conclusion

This cross-sectional study demonstrated that multiple factors, including experiences of medical decision-making, education level, knowledge of ACP, treatment expenditure, uncertainty of disease, social support, and life orientation, significantly impacted the engagement of SDMs in ACP. In conclusion, this study provided valuable insights into the factors influencing ACP participation among surrogate decision-makers for advanced cancer patients. Future research should further explore the specific mechanisms underlying these factors and develop targeted interventions to enhance ACP engagement of SDMs, ultimately improving the quality of end-of-life care for patients with advanced cancer.

## Supporting information

**S1 Table. Original data.**
(XLSX)

## Author contributions

**Data curation:** Zhihao Han.

**Formal analysis:** Zhihao Han.

**Investigation:** Guiyue Ma, Yong Fang, Linxia Tang.

**Supervision:** Xiaoqin Ma.

**Validation:** Jingjing Su.

**Writing – original draft:** Zhihao Han, Guiyue Ma.

**Writing – review & editing:** Jingjing Su, Xiaoqin Ma.

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
