## [Decision Letter · Decision Letter 0]

19 Nov 2024

PONE-D-24-35495Factors influencing the participation of surrogate decision-makers for advanced cancer patients in advance care planning: A cross-sectional studyPLOS ONE

Dear Dr. Ma,

Thank you for submitting your manuscript to PLOS ONE. After careful consideration, we feel that it has merit but does not fully meet PLOS ONE’s publication criteria as it currently stands. Therefore, we invite you to submit a revised version of the manuscript that addresses the points raised during the review process.

We look forward to receiving your revised manuscript.

Kind regards,

Xiong Xingyu, M.D.

Academic Editor

PLOS ONE

**Journal Requirements:**

**Additional Editor Comments:**

Actually, the peer reviewers have proposed many serious problems, and one of the experts recommended to reject your manuscript. After careful consideration, we decide to give authors a chance to comprehensively improve your paper by replying to all the reviewers' feedbacks. If these questions are properly addressed by authors and approved by reviewers, editors would like to reconsider the decision for publication.

Reviewers' comments:

Reviewer's Responses to Questions

**Comments to the Author**

1. Is the manuscript technically sound, and do the data support the conclusions?

Reviewer #1: No

Reviewer #2: Yes

Reviewer #3: Partly

2. Has the statistical analysis been performed appropriately and rigorously? 

Reviewer #1: I Don't Know

Reviewer #2: I Don't Know

Reviewer #3: I Don't Know

3. Have the authors made all data underlying the findings in their manuscript fully available?

Reviewer #1: Yes

Reviewer #2: Yes

Reviewer #3: No

4. Is the manuscript presented in an intelligible fashion and written in standard English?

Reviewer #1: No

Reviewer #2: Yes

Reviewer #3: No

5. Review Comments to the Author

**Reviewer #1: ** The authors have clearly identified surrogate decision-making as an important subject especially for patients with advanced cancer and the need for education and policy development in China. However, there are three main reasons why this article is not suitable for publication in its present form:

1) The English language is currently not up to publication standard and requires considerable editing. Also, there are incorrect facts in the Introduction about advance care planning in different countries, for example, Australia does not have federal laws regulating advance care planning, this is a Sate responsibility.

2) The research method does not address the key principle of surrogate decision-making, that is, what would the patient want, even though it is mentioned in the Introduction when defining Advance Care Planning, “… which aims to respect the patient’s values … “. There is no mention of this fundamental reason for surrogate decision making in the Aim or Methods. The authors note there are limitations with the research including no interviews, but do not address how the participants would know or understand the decisions the patient would want. This is the basic principle underpinning advance care planning and as the authors mention will require interviews to understand the complex issues involved.

3) Although the authors have not clearly explained their quantitative methodology, it is unclear how the participants were approached to be involved with this research. Were they given the questionnaire to complete, or were they assisted by one of the researchers. (As a qualitative methods research fellow, I have not commented on the quantitative results).

**Reviewer #2: ** Title: Factors influencing the participation of surrogate decision-makers for advanced cancer patients in advance care planning: A cross-sectional study.

Thank you for allowing me to review this important manuscript.

General feedback:

Overall, very well written manuscript and a well thought study.

Recommend using tenses consistently. Please have the manuscript reviewed for grammatical and sentence structure errors to bring more clarity to the content.

Recommend changing the term, “propaganda” as it has a negative connotation.

Abstract: Concise and well written. However, please identify the age group of patients you are focusing on.

Introduction: The flow of ideas is clear leading to your research question.

Methods and results:

Very thorough and well presented.

Discussion:

Please expand on your point regarding optimistic surrogate decision makers. Are you suggesting that people should be optimistic so healthcare providers can better engage them in ACP discussions? I think this is too much to ask from them when they are already dealing with cancer diagnosis and treatments. What can be done from the perspectives of healthcare providers and researchers to increase the uptake of ACP?

Conclusion:

Make conclusion concise and move future suggestions in research and practice implications as well as limitations to the discussion.

I recommend changing, “cognitive error” to another better term to describe your point.

**Reviewer #3: ** Comments:

Thank you for offering the opportunity to review the manuscript titled “Factors influencing the participation of surrogate decision-makers for advanced cancer patients in advance care planning: A cross-sectional study”.

I believe the dataset of this study have valuable and important figures that worth dissemination. However, the quality of the current manuscript is far below the standard ready to publish. Here is an overall comment to address:

1) Please justify why the chosen study population “surrogate decision-maker of patients with advanced cancer” is important to study in the Chinese context. There should be more elaboration about how the Chinese culture affecting medical decisions and the dynamic between family caregivers and doctors. This would highlight the significance of this study.

2) The statistical tests performed in inferential analyses were not clearly reported. It is unclear about how the missing data was treated, and the completion rate of each outcome measure. The result tables do not clearly describe if mean (SD) is reported or not.

3) The discussion is superficial to add new knowledge to the field of study. Consider comparing the results with the following papers:

- WANG, T., WANG, W., SHEN, W., & SHI, B. (2022). Participation in Advance Care Planning and Associated Factors among Surrogate Decision Makers of Patients with Hematologic Malignancies. Chinese General Practice, 25(07), 859.

- Liu, X., Chen, H., Zhang, L., Zhang, Q., Feng, T., & Liu, D. (2022). Advance care planning engagement among family members of community-dwelling elderly patients with chronic diseases in China: a mixed-methods study. Journal of Hospice & Palliative Nursing, 24(2), E26-E34.

Please refer to the attached PDF file marked with in-text comments for more specific comments about the methods, analysis, and results. Thank you!

6. PLOS authors have the option to publish the peer review history of their article (what does this mean? ). If published, this will include your full peer review and any attached files.

**Do you want your identity to be public for this peer review?** For information about this choice, including consent withdrawal, please see our Privacy Policy .

Reviewer #1: No

Reviewer #2: No

Reviewer #3: No

---

## [Author Response · Author response to Decision Letter 1]

2 Jan 2025

Reviewer #1: The authors have clearly identified surrogate decision-making as an important subject especially for patients with advanced cancer and the need for education and policy development in China. However, there are three main reasons why this article is not suitable for publication in its present form:

1) The English language is currently not up to publication standard and requires considerable editing. Also, there are incorrect facts in the Introduction about advance care planning in different countries, for example, Australia does not have federal laws regulating advance care planning, this is a Sate responsibility.

Response: Thank you for your comments. We have polished and revised the incorrected part in this paper.

2) The research method does not address the key principle of surrogate decision-making, that is, what would the patient want, even though it is mentioned in the Introduction when defining Advance Care Planning, “… which aims to respect the patient’s values … “. There is no mention of this fundamental reason for surrogate decision making in the Aim or Methods. The authors note there are limitations with the research including no interviews, but do not address how the participants would know or understand the decisions the patient would want. This is the basic principle underpinning advance care planning and as the authors mention will require interviews to understand the complex issues involved.

Response: Thank you for your comments. We have added the reason for surrogate decision making in the “Introduction” part. This study was mainly focus on the factors influencing the engagement of ACP, and we also invited some surrogate decision-makers to interview about the complex issues, but this part was not presented in this study.

3) Although the authors have not clearly explained their quantitative methodology, it is unclear how the participants were approached to be involved with this research. Were they given the questionnaire to complete, or were they assisted by one of the researchers. (As a qualitative methods research fellow, I have not commented on the quantitative results).

Response: Thank you for reminding us to add the methods of this study. We have added the procedure of our study in the revised manuscript in the “Study design” part.

Reviewer #2: Title: Factors influencing the participation of surrogate decision-makers for advanced cancer patients in advance care planning: A cross-sectional study.

Thank you for allowing me to review this important manuscript.

General feedback: Overall, very well written manuscript and a well thought study.

Recommend using tenses consistently. Please have the manuscript reviewed for grammatical and sentence structure errors to bring more clarity to the content.

Response: Thank you for reminding us the tenses in the paper, grammatical and sentence structure errors; and we have revised the full paper.

Recommend changing the term, “propaganda” as it has a negative connotation.

Response: Thank you for your comments. We have changed the term, “propaganda” into “dissemination”.

Abstract: Concise and well written. However, please identify the age group of patients you are focusing on.

Response: We mainly focused on the surrogate decision-makers of advanced patients, and patients who were eligible at different age groups were all recruited in this study.

Introduction: The flow of ideas is clear leading to your research question.

Methods and results: Very thorough and well presented.

Discussion: Please expand on your point regarding optimistic surrogate decision makers. Are you suggesting that people should be optimistic so healthcare providers can better engage them in ACP discussions? I think this is too much to ask from them when they are already dealing with cancer diagnosis and treatments. What can be done from the perspectives of healthcare providers and researchers to increase the uptake of ACP?

Response: Thank you for your comments and sorry for the misunderstanding. As your opinions, we could not ask them to do too much because they have already suffered a lot. However, as the study shown, the optimistic participants presented more active engagement of ACP, and it might be helpful for healthy caregivers to evaluate the status of patient or surrogate decision-makers to avoid some conflicts during the ACP.

Conclusion: Make conclusion concise and move future suggestions in research and practice implications as well as limitations to the discussion. I recommend changing, “cognitive error” to another better term to describe your point.

Response: Thank you for your comments. We have change the “cognitive error” into “misunderstanding”.

Reviewer #3: Comments: Thank you for offering the opportunity to review the manuscript titled “Factors influencing the participation of surrogate decision-makers for advanced cancer patients in advance care planning: A cross-sectional study”.

I believe the dataset of this study have valuable and important figures that worth dissemination. However, the quality of the current manuscript is far below the standard ready to publish. Here is an overall comment to address:

1) Please justify why the chosen study population “surrogate decision-maker of patients with advanced cancer” is important to study in the Chinese context. There should be more elaboration about how the Chinese culture affecting medical decisions and the dynamic between family caregivers and doctors. This would highlight the significance of this study.

Response: Thank you for your comments. We have added the related information in the introduction part.

2) The statistical tests performed in inferential analyses were not clearly reported. It is unclear about how the missing data was treated, and the completion rate of each outcome measure. The result tables do not clearly describe if mean (SD) is reported or not.

Response: Thank you for your comments. We have revised the Statistical Analysis part. “The socio-demographic and cancer characteristics of SDMs were described by frequency and percentage. The scores of questionnaires were described by mean ± standard deviation or median (quartile) for normal or non-normal distribution data”.

3) The discussion is superficial to add new knowledge to the field of study. Consider comparing the results with the following papers:

- WANG, T., WANG, W., SHEN, W., & SHI, B. (2022). Participation in Advance Care Planning and Associated Factors among Surrogate Decision Makers of Patients with Hematologic Malignancies. Chinese General Practice, 25(07), 859.

- Liu, X., Chen, H., Zhang, L., Zhang, Q., Feng, T., & Liu, D. (2022). Advance care planning engagement among family members of community-dwelling elderly patients with chronic diseases in China: a mixed-methods study. Journal of Hospice & Palliative Nursing, 24(2), E26-E34.

Response: Thank you for your suggestions. We have added those two papers in the discussion part.

Editor #1: I believe these three subscales should be reported in terms of mean (SD). Please specify.

Response: Thank you for the editor's suggestions. When we conducted the normality test on the data, we found that not all of the data conform to a normal distribution. Therefore, we only used mean (SD) to represent the data that meet the normal distribution criteria, while for the data that do not conform to a normal distribution, we used the median and interquartile range for representation. We have made revisions to the manuscript regarding the other issues mentioned by the editor.

---

## [Decision Letter · Decision Letter 1]

6 Feb 2025

PONE-D-24-35495R1Factors influencing the participation of surrogate decision-makers for advanced cancer patients in advance care planning: A cross-sectional studyPLOS ONE

Dear Dr. Ma,

Thank you for submitting your manuscript to PLOS ONE. After careful consideration, we feel that it has merit but does not fully meet PLOS ONE’s publication criteria as it currently stands. Therefore, we invite you to submit a revised version of the manuscript that addresses the points raised during the review process.

We look forward to receiving your revised manuscript.

Kind regards,

Xiong Xingyu, M.D.

Academic Editor

PLOS ONE

Reviewers' comments:

Reviewer's Responses to Questions

**Comments to the Author**

1. If the authors have adequately addressed your comments raised in a previous round of review and you feel that this manuscript is now acceptable for publication, you may indicate that here to bypass the “Comments to the Author” section, enter your conflict of interest statement in the “Confidential to Editor” section, and submit your "Accept" recommendation.

Reviewer #1: (No Response)

Reviewer #2: (No Response)

Reviewer #3: (No Response)

2. Is the manuscript technically sound, and do the data support the conclusions?

Reviewer #1: Partly

Reviewer #2: Partly

Reviewer #3: Partly

3. Has the statistical analysis been performed appropriately and rigorously? 

Reviewer #1: I Don't Know

Reviewer #2: I Don't Know

Reviewer #3: No

4. Have the authors made all data underlying the findings in their manuscript fully available?

Reviewer #1: No

Reviewer #2: Yes

Reviewer #3: Yes

5. Is the manuscript presented in an intelligible fashion and written in standard English?

Reviewer #1: No

Reviewer #2: No

Reviewer #3: No

6. Review Comments to the Author

Reviewer #1: The authors do not fully explore the principles behind advance care planning and substitute decision making. Their analysis largely ignores the fundamental issues of patient autonomy and of decision makers needing to implement the patient’s wishes. The authors mention that these are Western concepts that are new to China, but they do not make pay substantial attention to that point. The authors’ conclusions are sound, but the design of the study did not fully explore the points made in the conclusion. The research design was not the most appropriate design for this topic.

Reviewer #2: There are major areas that need revision to strengthen the manuscript for publication. These have been highlighted in the attached file.

Reviewer #3: The first and most crucial comment is to send your manuscript to professional editing before getting published. There were many grammatical mistakes and inconsistent use of terms, such as SDMs vs SMDs, severely affect readers’ understanding. More specific comments are as below:

Section 1: introduction

• The connection between ACP & AD was not highlighted, despite the two concepts were compared. Why promoting ACP is significant and how does it relate to the completion of AD? Please explain.

Section 2: Background

• Consider using the term “individual” instead of “patient” to minimize stigmatization.

Section 3: Aim

• Suggest moving the last sentence of this paragraph to the background section. It highlights the significance of the study rather than aim of this study.

Section 4.1: Study design

• Please state that this is a cross-sectional survey.

• Please clarify the duration of data collection period. It is not necessary to mention the period of evaluation (i.e. from July 2023 to January 2024).

• Please state the data collection approach. Whether participants were invited to complete a self-report questionnaire or assisted by research assistants.

Section 4.2: Participant selection

• Please clarify how SDM is appointed in your study setting. The phrase “appointed according to relevant regulations” is ambiguous.

Section 4.3: Measurements

• Please cite the references that support your choice of exploring factors under the three dimensions.

Section 4.3.3 to 4.4.5: Instruments

• The description of scoring methods was lengthy. Consider delete it as long as the total score is reported for analysis. Showing the direction of total score is adequate.

Section 4.4 Statistical analysis

• The term “cancer characteristics of SDMs” are ambiguous. The SDMs are not cancer patient. Do you mean “the characteristics of cancer of their care recipients”.

• Please state clearly if an independent t-test or paired t-test was used.

• When stating the p-value. Please consider if a small letter “p” or a capital letter “P” should be used.

• Consider addressing how missing data was treated in this section.

Section 5.2: Factors influencing the ACP engagement of SDMs

• This section reported the results of univariate analyses. The heading should be “factors associated with the ACP engagement of SDMs”

• The section heading should not be the same for section 5.2 and 5.3

Table 2:

• The heading of columns were not aligned with the heading of table. If subscale score is reported, please indicate these are the three subscale component of ACP engagement.

• In the column heading, please indicate “mean (SD)” was reported in the table.

• The choice of statistical tests for certain variables were queried. Please review the test being used for “marital status”, “experience of making medical decision”

• Please state the level of significance at the bottom of the table.

Table 3:

• Please clarify whether the construct reported here should be named as "life orientation" or "optimism"? Please align with section 4.3.5.

• What was the variable “attitudes” reported here? It was not described in section 4.3

Section 5.3

• The section heading should not be the same as section 5.2.

• Please check the presentation format of table 5. The collinearity statistics, TOL and VIF should be included in the table. Values for the constant is not necessarily to be reported.

• Please add the level of significance/ removal standard at the bottom of the table

Section 6: Discussion

• The conclusion “The univariate analyses showed that older SDMs had higher ACP engagement” was not supported by the statistics. From table 2, the mean ACP engagement score of the age group 18-40 is the highest among all. Please update your discussion.

• The discussion between educational level, cognitive ability and knowledge of ACP was not logical. People having higher educational level doesn’t mean they have higher cognitive ability, as they may suffer from cognitive problem such as memory problem. The discussion followed cannot address the need of the less educated group. Please revise the implication drawn from this finding.

• The discussion made based on the result of table 3 was not logical. Please explain why people having higher treatment expenditure reported higher level of ACP engagement; and why people having higher level of uncertainty of disease reported lower level of ACP engagement. Followed by the discussion of their implications.

• The recommendations made (public education and legal support) were not drawn from the study findings. Please tailored your recommendation specifically to a specific population that need extra attention and support to enhance their ACP engagement. And suggest what components should be included in the ACP intervention. Recommendations should be discussed in Section 6 instead of section 7.

• Please clarify the meaning of “validation studies”.

7. PLOS authors have the option to publish the peer review history of their article (what does this mean? ). If published, this will include your full peer review and any attached files.

**Do you want your identity to be public for this peer review?** For information about this choice, including consent withdrawal, please see our Privacy Policy .

Reviewer #1: No

Reviewer #2: No

Reviewer #3: No

---

## [Author Response · Author response to Decision Letter 2]

4 Mar 2025

Response to Reviewers

PONE-D-24-35495R1

Factors influencing the participation of surrogate decision-makers for advanced cancer patients in advance care planning: A cross-sectional study

We are very grateful to the reviewers for their revised comments on the manuscript, which are very important for the quality of the manuscript. On behalf of all the contributing authors, I would like to express our sincere appreciations of your letter and reviewers’ constructive comments concerning our article entitled “Factors influencing the participation of surrogate decision-makers for advanced cancer patients in advance care planning: A cross-sectional study”. We revised the manuscript again based on the reviewers’ comments, hoping to meet PLOS ONE’s publication criteria.

Reviewer #1: The authors do not fully explore the principles behind advance care planning and substitute decision making. Their analysis largely ignores the fundamental issues of patient autonomy and of decision makers needing to implement the patient’s wishes. The authors mention that these are Western concepts that are new to China, but they do not make pay substantial attention to that point. The authors’ conclusions are sound, but the design of the study did not fully explore the points made in the conclusion. The research design was not the most appropriate design for this topic.

Response: Many thanks to the reviewers for their constructive comments. There have been comprehensive studies on the autonomy of cancer patients in participating in ACP, and the focus group of our study is the surrogate decision-makers of cancer patients. We only conducted a questionnaire survey on the willingness and influencing factors of surrogate decision-makers before formally participating in ACP. The surrogate decision-makers did not actually participate in ACP. Because currently ACP has not been fully implemented in mainland China, and no relevant laws have been promulgated, but our research conclusions can provide some suggestions for law makers. In addition, ACP began to rise in countries such as Europe and the United States, and mainland China should learn from it. Currently, mainland China has very limited policies and laws regarding ACP. Therefore, we are unable to provide a substantive explanation of the implementation of ACP in mainland China.

In response to the issue mentioned by the reviewer that the research design is not suitable for this topic, we would like to explain that conducting quantitative research is the most important part of our research topic. We are currently conducting qualitative research on patients’ surrogate decision-makers, with the purpose of supplementing and explaining the quantitative research results. We hope that subsequent research results can also be submitted to the PLOS ONE and have opportunity to be published.

Reviewer #2: There are major areas that need revision to strengthen the manuscript for publication. These have been highlighted in the attached file.

1) Multiple grammatical errors, and flow of ideas, need to be reviewed and edited throughout the manuscript. Please have the manuscript reviewed for grammatical and sentence structure errors to bring more clarity to the content.

Response: We sincerely appreciate the reviewers’ insightful comments and constructive suggestions, which have significantly improved the quality of our manuscript. We have polished the grammar and sentence structure of the manuscript and hope that the revised manuscript will meet the publication requirements in terms of grammar and sentence structure.

2) Need to check the facts again, for example, the implementation of AD has been guaranteed in UK etc. This is mentioned in the introduction. However, it is not guaranteed or legally binding there due to multifactorial reasons including legal and practical. Kindly review for each country listed and remove “etc.” wherever possible in the manuscript.

Response: The authors are grateful for the valuable feedback provided by the reviewers, as it has greatly enhanced the clarity and rigor of this work. We misspelled the England into UK. We have checked and revised the issues raised by the reviewer.

3) Background:

• Last paragraph of this section talks a lack of existing research on ACP engagement by SDMs. Are you referring to the situation in China? Please specify and elaborate with current evidence to support the need for this study.

• Talk about relational aspect to caregiving in Asian cultures, and how does it impact involvement in ACP.

• It is unclear if authors are referring to the need for the SDMs to engage in ACP with their patients living with cancer or by themselves. Please clarify. ACP is not possible without the presence of the person in question.

Response: We thank the reviewers for their thoughtful critiques, which have guided us in refining both the methodology and presentation of results. We first explain the doubts raised by the reviewers. We then discussed the impact of Asian nursing cultural background on participation in ACP. In addition, we need to explain that ACP emphasizes the joint participation of patients and patient surrogate decision-makers. Therefore, the main purpose of our study is to understand the willingness and influencing factors of patients’ surrogate decision-makers to participate in the formulation and implementation of patients’ ACP, rather than to explore the surrogate decision-makers themselves’ participation in ACP. There are many studies focus on ACP engagement, while the study about ACP engagement of SDMs in mainland China is less, and this is the first large sample study focusing on the factors influencing ACP engagement of SDMs of different advanced cancer patients. We hope that our explanation will provide reviewers with a clear understanding of the purpose of this study.

4) Study Design:

• Clearly identify the study design used in this section e.g., cross sectional study.

Response: The constructive criticisms from the reviewers have been instrumental in strengthening the arguments and addressing gaps in the original submission. We provide a clear explanation of study design types in this section.

5) Participants and selection criteria:

• Consider using the term, “participants” instead of subjects.

• Move participant related information in this section from the study design section.

• Clearly identify who recruited participants and how e.g., nurse or doctor in the department. What was the role of investigators in recruitment. Procedure of recruitment remains unclear.

• Was the survey electronic or paper-based.

Response: We deeply acknowledge the reviewers’ expertise, and their detailed suggestions have led to a more robust and coherent analysis. We have replaced “subjects” with “participants” based on comments made by reviewers. We have moved participant related information in this section from the study design section. Our investigators are hospital nurses and the questionnaire is in the form of a paper questionnaire. The investigators were nurses in the department, and completed all the recruitment.

6) Measurements:

• Please cite the studies utilized for devising sociodemographic survey.

Response: We extend our gratitude to the reviewers for their meticulous evaluation. We have added relevant references where necessary in the manuscript.

7) Results:

• How was “gender” captured, did you only include female and male? These are biological constructs whereas, gender is a social construct. Please review and change accordingly.

Response: We sincerely appreciate the reviewer’s insightful comment regarding the distinction between biological sex and gender as a social construct. In response to this concern, we have replaced the term “gender” with “sex” throughout the manuscript.

8) Discussion:

• Your point about treatment expenditure and ACP is not clear. What are the clear recommendations here? Do you think people with less might be coerced into ACP discussions?

Response: Thank you for your valuable feedback. We understand your concern about the clarity of the relationship between treatment expenditure and advance care planning (ACP) and the potential implications of economic pressure on decision - making.

In our study, we found that high treatment costs significantly increased the economic burden on families, which in turn made them more inclined to consider less aggressive medical care measures. This economic pressure led to a higher willingness to participate in ACP discussions. It’s important to note that this does not necessarily mean that people with less economic means were coerced into ACP discussions. Rather, the financial strain influenced their preferences and considerations regarding end - of - life care options.

To address your concerns and provide clearer recommendations, we suggest that healthcare providers should be aware of the economic impact on patients and their families when discussing treatment options and ACP. They should offer support and guidance to help patients and their families make informed decisions that align with their values and financial situations. Additionally, policymakers could consider measures to reduce the financial burden of cancer treatment, such as improving insurance coverage and providing financial assistance programs, to ensure that patients have equal access to appropriate care and are not unduly influenced by economic factors in ACP discussions.

We have revised the manuscript to better clarify these points and provide more detailed recommendations. We hope this addresses your concerns and enhances the quality of our research.

• Throughout your discussion, you don’t talk about the central role of the patient in ACP. ACP can’t happen without the active involvement of the patient. What are your recommendations to bring patients and SDM together to involve in ACP.

Response: Thank you for your insightful comment. You are right that the patient’s central role in advance care planning (ACP) is crucial. In our study, we focused on the factors affecting the involvement of SDMs. However, we acknowledge that we did not sufficiently emphasize the patient’s active involvement in ACP. To address your concern and provide clear recommendations on how to bring patients and SDMs together in ACP, we suggest the following:

Patients play a central role in the development and implementation of the entire ACP, and it is crucial to facilitate communication between patients and SDMs. Healthcare providers should play an active role in facilitating communication between patients and SDMs. This includes providing clear and comprehensive information about ACP, its benefits, and the importance of the patient’s active participation. Educational sessions and materials can help both patients and SDMs understand the process and their respective roles. Establishing a trusting relationship between patients, SDMs, and healthcare providers is essential. This can be achieved through open and empathetic communication, where patients feel comfortable expressing their wishes and concerns. Each patient and family is unique, and the approach to ACP should be tailored to their specific needs and circumstances. Healthcare providers should be sensitive to cultural, religious, and personal beliefs and preferences, and adapt their communication and support accordingly.

We have revised our manuscript to better address the patient’s central role in ACP and to provide more detailed recommendations on how to involve both patients and SDMs in the process. We hope these changes address your concerns and enhance the quality of our research.

• What is your opinion on goals of care discussion for patients who are not able to participate in ACP process?

Response: Thank you for raising this important issue. You are correct that for patients who are not able to participate in the ACP process, the goals of care discussion become even more critical. In our study, titled "Factors influencing the participation of surrogate decision - makers for advanced cancer patients in advance care planning: A cross - sectional study," we focused on the factors affecting the involvement of SDMs. However, we acknowledge that we did not sufficiently address the specific needs of patients who cannot participate in ACP.

We should enhance communication between patients and surrogate decision-makers and emphasize the guiding role that healthcare providers play in this. We have added relevant content in the discussion section. We hope these changes address your concerns and enhance the quality of our research.

• Future implications for practice and research need to be further elaborated.

Response: Thank you for your valuable feedback. We appreciate your suggestion that the future implications for practice and research need to be further elaborated. In response to this, we have revised our manuscript to include more detailed discussions on the future directions for both clinical practice and research.

Our study highlights the importance of SDMs in ACP for advanced cancer patients. Future practice should focus on providing healthcare providers with enhanced training on how to effectively communicate with SDMs and patients about ACP. This includes training on cultural sensitivity, communication techniques, and strategies to overcome barriers to ACP discussions. ACP should be integrated into routine care for advanced cancer patients. Healthcare providers should make it a standard practice to discuss ACP with patients and their SDMs early in the disease trajectory. This can help ensure that patients’ preferences and values are respected and that their care aligns with their wishes. In addition, future research should focus on longitudinal studies to better understand the long - term impact of ACP on patient outcomes and family satisfaction. Meanwhile, there is a need for more intervention studies to test the effectiveness of different ACP strategies.

We have revised our manuscript to include these points in the discussion section, providing a more comprehensive outlook on the future implications of our study. We believe these additions will enhance the value of our research and provide a clearer direction for future practice and research.

9) Limitations:

• Please elaborate on your point about “individual perspectives of participants were not included.”

Response: Thank you for your insightful comment. We apologize for not elaborating on the point about "individual perspectives of participants were not included" in the original manuscript. Here’s a more detailed explanation:

In this study, we mainly focused on the objective data and collective patterns. The individual perspectives, such as the outlooks of world, life and value, were indeed not taken into account. This is because the research design was primarily aimed at exploring the general trends and relationships among the variables at a macro level. The inclusion of individual perspectives would have required a more in-depth and qualitative approach, which was beyond the scope of the current research. However, we fully acknowledge that this limitation has affected the comprehensiveness of the study. We will make up for this shortcoming in subsequent research. We have also suggested that future research could adopt a mixed - methods approach to incorporate both the objective data and the individual perspectives, so as to provide a more comprehensive understanding of the issue.

Once again, we appreciate your valuable suggestion and we have made the corresponding revisions to improve the quality of the paper.

Reviewer #3: The first and most crucial comment is to send your manuscript to professional editing before getting published. There were many grammatical mistakes and inconsistent use of terms, such as SDMs vs SMDs, severely affect readers’ understanding. More specific comments are as below.

1) Section 1: introduction

• The connection between ACP & AD was not highlighted, despite the two concepts were compared. Why promoting ACP is significant and how does it relate to the completion of AD? Please explain.

Response: Thank you for your insightful comment regarding the connection between Advance Care Planning (ACP) and Advance Directives (AD). We appreciate this opportunity to clarify and elaborate on the si

---

## [Decision Letter · Decision Letter 2]

26 Mar 2025

PONE-D-24-35495R2Factors influencing the participation of surrogate decision-makers for advanced cancer patients in advance care planning: A cross-sectional studyPLOS ONE

Dear Dr. Ma,

Thank you for submitting your manuscript to PLOS ONE. After careful consideration, we feel that it has merit but does not fully meet PLOS ONE’s publication criteria as it currently stands. Therefore, we invite you to submit a revised version of the manuscript that addresses the points raised during the review process.

We look forward to receiving your revised manuscript.

Kind regards,

Xiong Xingyu, M.D.

Academic Editor

PLOS ONE

Additional Editor Comments:

Peer experts still point out many serious problems, and there is always a rejection recommendation in each round of peer review. Especially, language problems were proposed by each reviewer. Thus, this revision decision is the last chance to improve you manuscript for potential publication in PLOS ONE. I hope your next ammendments can meet the standard of the experts. I look forward to your revisions. Thanks for the chance to consider your work.

Reviewers' comments:

Reviewer's Responses to Questions

**Comments to the Author**

1. If the authors have adequately addressed your comments raised in a previous round of review and you feel that this manuscript is now acceptable for publication, you may indicate that here to bypass the “Comments to the Author” section, enter your conflict of interest statement in the “Confidential to Editor” section, and submit your "Accept" recommendation.

Reviewer #2: (No Response)

Reviewer #3: (No Response)

2. Is the manuscript technically sound, and do the data support the conclusions?

Reviewer #2: Partly

Reviewer #3: Partly

3. Has the statistical analysis been performed appropriately and rigorously? 

Reviewer #2: Yes

Reviewer #3: Yes

4. Have the authors made all data underlying the findings in their manuscript fully available?

Reviewer #2: Yes

Reviewer #3: Yes

5. Is the manuscript presented in an intelligible fashion and written in standard English?

Reviewer #2: No

Reviewer #3: No

6. Review Comments to the Author

Reviewer #2: Review

Title: Factors influencing the participation of surrogate decision-makers for advanced cancer patients in advance care planning: A cross-sectional study.

General Comments

- Thank you for allowing me to review this manuscript for the third time. Most of the comments have been responded to however there are still some points that need addressing.

- Multiple grammatical errors, and flow of ideas, need to be reviewed and edited throughout the manuscript. Please have the manuscript reviewed for grammatical and sentence structure errors to bring more clarity to the content.

Major Comments

Background:

- Last paragraph of this section talks a lack of existing research on ACP engagement by SDMs. But it is still unclear if authors are referring to the need for the SDMs to engage in ACP with or without their patients (family members).

- It is unclear how will you use the findings of this study apart from education. Who will be the target for the education

- You have mentioned education, but what is your opinion on reducing/addressing the hierarchy existing between elders and physicians. How will knowing SDMs’ perspective help with removing barriers related to hierarchy?

Discussion:

- Your rationalization on yours and Liu study differences are not very clear.

- Its unclear what you mean by “new things” as a prerequisite for participation in ACP, please elaborate.

- You talk about the role of government, media and hospital in establishing theoretical framework. I don’t believe that’s their role. But I am also wondering if you are referring to policies instead of theoretical framework.

- Your point regarding higher financial burden leading to high participation in ACP by SDMs is contradictory to the actual philosophy of ACP, which must involve the patient. It appears you are referring to care decisions made by SDMs only.

- You talk about patients playing a central role in developing ACP and suggest healthcare providers should be sensitive to culture, religion etc., these are generic suggestions. I am interested to know recommendations related to the Mainland China.

Reviewer #3: The authors' effort in improving the writing is well acknowledged. Despite two rounds of extensive review from multiple reviewers, the article is still not ready to be published. First, the academic writing skills were beyond standard, which highly affected readers' understanding. More than one reviewer requested professional editing to improve clarity, yet the recommendation has not been taken. Secondly, inconsistent information was noted; careful proof-reading is required. The discussion fairly brings out new insights into the subject matter, as the discussion is rather superficial or lacks logic. Some suggestions or implications of the study were not supported by the findings. Please refer to the attachment for detailed comments.

7. PLOS authors have the option to publish the peer review history of their article (what does this mean? ). If published, this will include your full peer review and any attached files.

**Do you want your identity to be public for this peer review?** For information about this choice, including consent withdrawal, please see our Privacy Policy .

Reviewer #2: No

Reviewer #3: No

---

## [Author Response · Author response to Decision Letter 3]

16 Apr 2025

Response to Reviewers

Factors influencing the participation of surrogate decision-makers for advanced cancer patients in advance care planning: A cross-sectional study

Thank you for providing us with this final opportunity to revise our manuscript and for your continued guidance throughout the review process. We sincerely appreciate the detailed feedback from the reviewers and the editorial team, and we have carefully addressed all concerns raised in the previous decision letter. Attached please find the revised manuscript with tracked changes, along with a point-by-point response to each comment. We hope these revisions now fully meet the journal’s requirements and look forward to your final evaluation.

Reviewer:

1) Multiple grammatical errors, and flow of ideas, need to be reviewed and edited throughout the manuscript. Please have the manuscript reviewed for grammatical and sentence structure errors to bring more clarity to the content.

Response: Regarding your first comment about grammatical errors and the flow of ideas, we have taken immediate action to address this issue.

We have engaged a professional editing service to thoroughly review and polish the manuscript, ensuring that all grammatical and sentence structure errors have been corrected. In fact, we previously engaged a language editing service to assist with manuscript revisions, and we have again requested their assistance to polish the language of the manuscript for this submission. This process has significantly improved the clarity and readability of the content, and we are confident that the manuscript now meets the journal's requirements for language quality.

Additionally, we have attached a certificate from the editing service as proof of the language polishing. Please find it in the attachment for your reference.

2) Background: Last paragraph of this section talks a lack of existing research on ACP engagement by SDMs. But it is still unclear if authors are referring to the need for the SDMs to engage in ACP with or without their patients (family members).

Response: Regarding your comment about the lack of existing research on ACP engagement by SDMs, we would like to clarify:

The manuscript emphasizes the insufficient research on SDMs engagement in ACP with patients, rather than SDMs engagement in ACP without patients. We have revised the relevant section to better reflect this focus and to avoid any potential misunderstanding.

We sincerely thank you for pointing out this issue, which has helped us improve the clarity and accuracy of our manuscript.

3) Background: It is unclear how will you use the findings of this study apart from education. Who will be the target for the education.

Response: Thank you for your insightful comments on our manuscript. We have carefully considered your feedback and made the following revisions:

①Clarified the application of the study findings beyond just education. We have specified that the educational scheme will target SDMs of advanced cancer patients and aim to enhance their understanding and participation in ACP.

②Specified the target audience for the education as SDMs of advanced cancer patients.

We believe these revisions address your concerns and strengthen the manuscript. We appreciate your time and effort in reviewing our work and providing valuable feedback.

4) Background: You have mentioned education, but what is your opinion on reducing/addressing the hierarchy existing between elders and physicians. How will knowing SDMs’ perspective help with removing barriers related to hierarchy?

Response: Thank you for your insightful comments on our manuscript. We have carefully considered your feedback and made the following revisions:

Addressed the issue of hierarchy by discussing how understanding SDMs' perspectives can help identify and reduce barriers related to the hierarchical respect for elders and physicians. We have emphasized the importance of promoting more egalitarian communication and decision-making processes through targeted education.

We believe these revisions address your concerns and strengthen the manuscript.

5) Discussion: Your rationalization on yours and Liu study differences are not very clear.

Response: Thank you for your insightful comments on our manuscript. We have carefully considered your feedback and made the following revisions to address the differences between our study and Liu's study:

①Clarified the specific characteristics of the care recipients in both studies, highlighting that Liu's study focused on elderly patients with chronic diseases, while our study centered on advanced cancer patients.

②Provided a more detailed explanation of why younger SDMs in our study exhibited higher ACP engagement, considering factors such as adaptability and access to resources.

We believe these revisions provide a clearer rationale for the differences observed in our study compared to Liu's study and strengthen the manuscript. We appreciate your time and effort in reviewing our work and providing valuable feedback.

6) Discussion: Its unclear what you mean by "new things" as a prerequisite for participation in ACP, please elaborate.

Response: Thank you for your insightful comments on our manuscript. We have carefully considered your feedback and made the following revisions:

We agree that the term "new things" required clarification. In the revised manuscript, we have replaced it with "relatively unfamiliar healthcare concepts like ACP" and added contextual explanation about cultural barriers to end-of-life discussions. This modification better specifies that ACP itself represents a novel decision-making paradigm in certain sociocultural settings.

We appreciate your feedback in strengthening the precision of this critical conceptual linkage.

7) Discussion: You talk about the role of government, media and hospital in establishing theoretical framework. I don’t believe that’s their role. But I am also wondering if you are referring to policies instead of theoretical framework.

Response: Thank you for your insightful comments on our manuscript. We have carefully considered your feedback.

We acknowledge that the original phrasing regarding "theoretical frameworks" was ambiguous and potentially misleading. As you rightly pointed out, these entities are better positioned to implement practical policies rather than abstract theoretical constructs. In the revised text, we have replaced "theoretical frameworks" with "policy-driven strategies" to better reflect the practical roles of governments (e.g., funding policies and regulatory support), media (e.g., public awareness initiatives), and hospitals (e.g., integrating ACP into clinical workflows).

We sincerely appreciate your expertise in identifying this inconsistency.

8) Discussion: Your point regarding higher financial burden leading to high participation in ACP by SDMs is contradictory to the actual philosophy of ACP, which must involve the patient. It appears you are referring to care decisions made by SDMs only.

Response: Thank you for your astute observation. We acknowledge that the original phrasing could inadvertently imply that SDMs’ financial motivations replace patient-centered ACP principles, which contradicts the core ethos of shared decision-making. In the revised text, we have:

①Clarified that SDMs’ cost-related decisions arise only when patient preferences are undocumented, emphasizing ACP’s foundational goal of proactively capturing patient autonomy.

②Reframed the financial burden as a catalyst for advocating early patient engagement in ACP, aligning with its philosophy of preemptive communication.

Your critique has strengthened the manuscript’s alignment with ACP’s ethical framework. We deeply appreciate your expertise.

9) Discussion: You talk about patients playing a central role in developing ACP and suggest healthcare providers should be sensitive to culture, religion etc., these are generic suggestions. I am interested to know recommendations related to the Mainland China.

Response: Thank you for emphasizing the need for region-specific insights. We have substantially revised this section to address Mainland China’s unique challenges and opportunities in ACP implementation:

①Added evidence on SDMs’ predominant role in family hierarchies, citing studies about filial piety’s dual impact on ACP engagement.

② Researchers could develop ACP decision aids tailored to Chinese culture. These tools should primarily focus on patient-related information, helping patients understand their disease characteristics and predict treatment outcomes. Currently, there are no researchers in Mainland China who have developed relevant decision aids.

③ Aligned hospital policies with the "Healthy China 2030" framework, emphasizing measurable outcomes (e.g., ACP documentation rates) to drive institutional change.

These additions, supported by region-specific references [47], transform generic recommendations into actionable strategies for the Chinese context. We deeply appreciate your guidance in enhancing the manuscript’s relevance.

Thank you once again for your patience and constructive feedback. We have dedicated significant effort to refining this manuscript in response to the reviewers’ and editors’ concerns. Should any further revisions be required, please do not hesitate to contact us. We sincerely hope that the current version now aligns with the journal’s standards and would be honoured to contribute to PLOS ONE’s scientific community.

Best regards,

Zhihao Han, Jingjing Su, Guiyue Ma, Yong Fang, Linxia Tang, Xiaoqin Ma

Zhejiang Chinese Medical University

20041028@zcmu.edu.cn

---

## [Decision Letter · Decision Letter 3]

23 Apr 2025

PONE-D-24-35495R3Factors influencing the participation of surrogate decision-makers for advanced cancer patients in advance care planning: A cross-sectional studyPLOS ONE

Dear Dr. Ma,

Thank you for submitting your manuscript to PLOS ONE. After careful consideration, we feel that it has merit but does not fully meet PLOS ONE’s publication criteria as it currently stands. Therefore, we invite you to submit a revised version of the manuscript that addresses the points raised during the review process.

We look forward to receiving your revised manuscript.

Kind regards,

Xiong Xingyu, M.D.

Academic Editor

PLOS ONE

**Journal Requirements:**

Reviewers' comments:

Reviewer's Responses to Questions

**Comments to the Author**

1. If the authors have adequately addressed your comments raised in a previous round of review and you feel that this manuscript is now acceptable for publication, you may indicate that here to bypass the “Comments to the Author” section, enter your conflict of interest statement in the “Confidential to Editor” section, and submit your "Accept" recommendation.

Reviewer #2: (No Response)

Reviewer #3: All comments have been addressed

2. Is the manuscript technically sound, and do the data support the conclusions?

Reviewer #2: Yes

Reviewer #3: Yes

3. Has the statistical analysis been performed appropriately and rigorously? 

Reviewer #2: Yes

Reviewer #3: Yes

4. Have the authors made all data underlying the findings in their manuscript fully available?

Reviewer #2: Yes

Reviewer #3: Yes

5. Is the manuscript presented in an intelligible fashion and written in standard English?

Reviewer #2: Yes

Reviewer #3: Yes

6. Review Comments to the Author

**Reviewer #2:**  Authors have satisfactorily responded to the reviewers' comments and hence can be considered for publication.

**Reviewer #3: ** 1. Background

In the 4th paragraph, I believe the meaning of “there are few studies focusing on the ACP engagement of SDMs with patients in mainland China” is deviated from your original meaning. Please consider rewrite it as “there are few studies related to ACP involving both SDMs and patients in mainland China.

In the last paragraph of section 2, the elaboration on literature gap closely resembles the discussion section, which sounds duplicated and a preconception before investigation.

To address the literature gap, please consider citing the following reference to highlight the complexity of implementing ACP in mainland China.

McMahan, R. D., Tellez, I., & Sudore, R. L. (2021). Deconstructing the Complexities of Advance Care Planning Outcomes: What Do We Know and Where Do We Go? A Scoping Review. Journal of the American Geriatrics Society, 69(1), 234–244. https://doi.org/10.1111/jgs.16801

2. Methods

In section 4.1 and 4.2, the handling of missing data has been reported in section 4.4. Please delete it from here.

3. Results

In section 5.1, according to the figure in table 1, only 8.7% of advanced cancer patients were fully independent.

Please clarify if the author would like to emphasize that over 90% of them require ADL support? It sounds more logical to emphasize the caregiver burden in this way than emphasizing that two-thirds of them could take care of themselves.

4. Discussion

In the 7th paragraph, please consider citing references to support the first two recommendations (involving SDMs in ACP and developing decision aids), especially literature related to the Chinese cultural context:

Yeung, C. C. Y., Ho, K. H. M., & Chan, H. Y. L. (2023). A dyadic advance care planning intervention for people with early-stage dementia and their family caregivers in a community care setting: a feasibility trial. BMC geriatrics, 23(1), 115.

In the 8th pagraph, please consider citing the following reference to support the importance of healthcare provider training:

Mills, J., Kim, S. H., Chan, H. Y., Ho, M. H., Montayre, J., Liu, M. F., & Lin, C. C. (2021). Palliative care education in the Asia Pacific: challenges and progress towards palliative care development. Progress in Palliative Care, 29(5), 251-254.

7. PLOS authors have the option to publish the peer review history of their article (what does this mean? ). If published, this will include your full peer review and any attached files.

**Do you want your identity to be public for this peer review?** For information about this choice, including consent withdrawal, please see our Privacy Policy .

Reviewer #2: No

Reviewer #3: No

---

## [Author Response · Author response to Decision Letter 4]

3 May 2025

Factors influencing the participation of surrogate decision-makers for advanced cancer patients in advance care planning: A cross-sectional study

We sincerely appreciate the detailed feedback from the reviewers and the editorial team, and we have carefully addressed all concerns raised in the decision letter. Attached please find the revised manuscript with tracked changes, along with a point-by-point response to each comment. We hope these revisions now fully meet the journal’s requirements and look forward to your final evaluation.

Reviewer:

1) Background: In the 4th paragraph, I believe the meaning of “there are few studies focusing on the ACP engagement of SDMs with patients in mainland China” is deviated from your original meaning. Please consider rewrite it as “there are few studies related to ACP involving both SDMs and patients in mainland China.

Response: We would like to express our sincere gratitude for your valuable comments and suggestions on our manuscript. We have carefully considered your feedback and made the necessary revisions.

Regarding the statement in the 4th paragraph, we agree with your assessment that the original wording may have deviated from our intended meaning. We have revised the sentence to “there are few studies related to ACP involving both SDMs and patients in mainland China” as you suggested. We believe this revision more accurately conveys our original intention and provides a clearer understanding of the research landscape in this area.

Once again, we thank you for your time and insightful comments. We are confident that these revisions have improved the quality of our manuscript and we look forward to the opportunity to address any further feedback you may have.

2) In the last paragraph of section 2, the elaboration on literature gap closely resembles the discussion section, which sounds duplicated and a preconception before investigation.

Response: Thank you for your valuable feedback on our manuscript. We've carefully reviewed your comments and made the appropriate revisions.

We acknowledge your observation regarding the overlap between the elaboration on the literature gap in the last paragraph of Section 2 and the discussion section, which could give an impression of duplication and preconception. In response, we have removed the redundant content from Section 2 to ensure each section has a distinct focus and contributes uniquely to the manuscript.

We believe these revisions have improved the clarity and logical flow of our paper, and we appreciate your time and insightful comments.

3) To address the literature gap, please consider citing the following reference to highlight the complexity of implementing ACP in mainland China.

McMahan, R. D., Tellez, I., & Sudore, R. L. (2021). Deconstructing the Complexities of Advance Care Planning Outcomes: What Do We Know and Where Do We Go? A Scoping Review. Journal of the American Geriatrics Society, 69(1), 234–244. https://doi.org/10.1111/jgs.16801

Response: Thank you for your insightful comments and suggestions on our manuscript. We have carefully considered your feedback and made the appropriate revisions.

In response to your suggestion to address the literature gap, we have added the reference you recommended. We believe this addition strengthens our analysis and provides a more comprehensive perspective on the challenges and intricacies of ACP in this specific cultural context.

Thank you again for your time and valuable insights. We are confident that these revisions have improved the quality of our manuscript and we look forward to the opportunity to address any further feedback you may have.

4) Methods: In section 4.1 and 4.2, the handling of missing data has been reported in section 4.4. Please delete it from here.

Response: Thank you for your valuable feedback on our manuscript. We have carefully reviewed your comments and made the necessary revisions.

In response to your suggestion regarding the methods section, we have removed the mention of missing data handling from sections 4.1 and 4.2.

We appreciate your guidance in helping us improve the clarity and organization of our manuscript. Thank you again for your time and insights.

5) Results: In section 5.1, according to the figure in table 1, only 8.7% of advanced cancer patients were fully independent.

Response: Thank you for bringing this to our attention. In response to your feedback regarding section 5.1, we have revised the description to more accurately reflect the data presented in table 1. The original statement has been updated to emphasize the minority of fully independent patients and clarify the distribution of dependency levels among the majority.

Thank you for your guidance in enhancing the precision of our manuscript.

6) Please clarify if the author would like to emphasize that over 90% of them require ADL support? It sounds more logical to emphasize the caregiver burden in this way than emphasizing that two-thirds of them could take care of themselves.

Response: Thank you for your insightful feedback. In response to your suggestion, we have revised the description in section 5.1 to better emphasize the caregiver burden. The original statement has been updated to highlight that over 90% of patients require some level of ADL support, which more accurately reflects the data in table 1 and aligns with the intended emphasis on the challenges faced by caregivers.

Thank you again for your guidance in improving the clarity and logical flow of our manuscript.

7) Discussion: In the 7th paragraph, please consider citing references to support the first two recommendations (involving SDMs in ACP and developing decision aids), especially literature related to the Chinese cultural context:

Yeung, C. C. Y., Ho, K. H. M., & Chan, H. Y. L. (2023). A dyadic advance care planning intervention for people with early-stage dementia and their family caregivers in a community care setting: a feasibility trial. BMC geriatrics, 23(1), 115.

Response: Thank you for your valuable suggestions. We have carefully considered your comments and made the appropriate revisions to our manuscript.

In the 7th paragraph of the Discussion section, we have added the reference you suggested to support the first two recommendations. This addition provides a more solid foundation for our recommendations and aligns with the cultural context highlighted in your review.

We believe this revision strengthens our discussion and provides a more comprehensive perspective on the topic. Thank you again for your time and insightful comments.

8) In the 8th pagraph, please consider citing the following reference to support the importance of healthcare provider training:

Mills, J., Kim, S. H., Chan, H. Y., Ho, M. H., Montayre, J., Liu, M. F., & Lin, C. C. (2021). Palliative care education in the Asia Pacific: challenges and progress towards palliative care development. Progress in Palliative Care, 29(5), 251-254.

Response: Thank you for your valuable suggestion. In response to your request in the 8th paragraph, we have added the reference to Mills et al. (2021) to support the importance of healthcare provider training. This addition reinforces the significance of education in palliative care and provides a stronger foundation for our discussion.

Thank you again for your guidance in enhancing the scholarly rigor of our manuscript.

Thank you once again for your patience and constructive feedback. We have dedicated significant effort to refining this manuscript in response to the reviewers’ and editors’ concerns. Should any further revisions be required, please do not hesitate to contact us. We sincerely hope that the current version now aligns with the journal’s standards and would be honoured to contribute to PLOS ONE’s scientific community.

Best regards,

Zhihao Han, Jingjing Su, Guiyue Ma, Yong Fang, Linxia Tang, Xiaoqin Ma

Zhejiang Chinese Medical University

20041028@zcmu.edu.cn

---

## [Decision Letter · Decision Letter 4]

14 May 2025

Factors influencing the participation of surrogate decision-makers for advanced cancer patients in advance care planning: A cross-sectional study

PONE-D-24-35495R4

Dear Dr. Ma,

We’re pleased to inform you that your manuscript has been judged scientifically suitable for publication and will be formally accepted for publication once it meets all outstanding technical requirements.

Kind regards,

Xiong Xingyu, M.D.

Academic Editor

PLOS ONE

Additional Editor Comments (optional):

Thanks for the authors' efforts to comprehensively improve your manuscript according to editor's and reviewers' comments. I am pleased to inform you that your paper can be accepted for publication now. Thanks for the chance to assess your interesting and important work. Additionally, many thanks for all the reviewers' precious inputs.

Reviewers' comments:

Reviewer's Responses to Questions

**Comments to the Author**

1. If the authors have adequately addressed your comments raised in a previous round of review and you feel that this manuscript is now acceptable for publication, you may indicate that here to bypass the “Comments to the Author” section, enter your conflict of interest statement in the “Confidential to Editor” section, and submit your "Accept" recommendation.

Reviewer #3: All comments have been addressed

2. Is the manuscript technically sound, and do the data support the conclusions?

Reviewer #3: Yes

3. Has the statistical analysis been performed appropriately and rigorously? 

Reviewer #3: Yes

4. Have the authors made all data underlying the findings in their manuscript fully available?

Reviewer #3: No

5. Is the manuscript presented in an intelligible fashion and written in standard English?

Reviewer #3: Yes

6. Review Comments to the Author

Reviewer #3: Thank you for having me in this review exercise. Your efforts in addressing the comments are much appreciated.

7. PLOS authors have the option to publish the peer review history of their article (what does this mean? ). If published, this will include your full peer review and any attached files.

**Do you want your identity to be public for this peer review?** For information about this choice, including consent withdrawal, please see our Privacy Policy .

Reviewer #3: No

---

## [Editor Report · Acceptance letter]

PONE-D-24-35495R4

PLOS ONE

Dear Dr. Ma,

I'm pleased to inform you that your manuscript has been deemed suitable for publication in PLOS ONE. Congratulations! Your manuscript is now being handed over to our production team.

Kind regards,

on behalf of

Dr. Xiong Xingyu

Academic Editor

PLOS ONE